# LEARNING TO SOLVE THE CREDIT ASSIGNMENT PROBLEM

**Benjamin James Lansdell**
Department of Bioengineering
University of Pennsylvania
Pennsylvania, PA 19104
lansdell@seas.upenn.edu

**Prashanth Ravi Prakash**
Department of Bioengineering
University of Pennsylvania
Pennsylvania, PA 19104

**Konrad Paul Kording**
Department of Bioengineering
University of Pennsylvania
Pennsylvania, PA 19104

## ABSTRACT

Backpropagation is driving today's artificial neural networks (ANNs). However, despite extensive research, it remains unclear if the brain implements this algorithm. Among neuroscientists, reinforcement learning (RL) algorithms are often seen as a realistic alternative: neurons can randomly introduce change, and use unspecific feedback signals to observe their effect on the cost and thus approximate their gradient. However, the convergence rate of such learning scales poorly with the number of involved neurons. Here we propose a hybrid learning approach. Each neuron uses an RL-type strategy to learn how to approximate the gradients that backpropagation would provide. We provide proof that our approach converges to the true gradient for certain classes of networks. In both feedforward and convolutional networks, we empirically show that our approach learns to approximate the gradient, and can match or the performance of exact gradient-based learning. Learning feedback weights provides a biologically plausible mechanism of achieving good performance, without the need for precise, pre-specified learning rules.

## 1 INTRODUCTION

It is unknown how the brain solves the credit assignment problem when learning: how does each neuron know its role in a positive (or negative) outcome, and thus know how to change its activity to perform better next time? This is a challenge for models of learning in the brain.

Biologically plausible solutions to credit assignment include those based on reinforcement learning (RL) algorithms and reward-modulated STDP (Bouvier et al., 2016; Fiete et al., 2007; Fiete & Seung, 2006; Legenstein et al., 2010; Miconi, 2017). In these approaches a globally distributed reward signal provides feedback to all neurons in a network. Essentially, changes in rewards from a baseline, or expected, level are correlated with noise in neural activity, allowing a stochastic approximation of the gradient to be computed. However these methods have not been demonstrated to operate at scale. For instance, variance in the REINFORCE estimator (Williams, 1992) scales with the number of units in the network (Rezende et al., 2014). This drives the hypothesis that learning in the brain must rely on additional structures beyond a global reward signal.

In artificial neural networks (ANNs), credit assignment is performed with gradient-based methods computed through backpropagation (Rumelhart et al., 1986; Werbos, 1982; Linnainmaa, 1976). This is significantly more efficient than RL-based algorithms, with ANNs now matching or surpassing human-level performance in a number of domains (Mnih et al., 2015; Silver et al., 2017; LeCun et al., 2015; He et al., 2015; Haenssle et al., 2018; Russakovsky et al., 2015). However there are well known problems with implementing backpropagation in biologically realistic neural networks.

One problem is known as weight transport (Grossberg, 1987): an exact implementation of back-propagation requires a feedback structure with the same weights as the feedforward network to communicate gradients. Such a symmetric feedback structure has not been observed in biological neural circuits. Despite such issues, backpropagation is the only method known to solve supervised and reinforcement learning problems at scale. Thus modifications or approximations to backpropagation that are more plausible have been the focus of significant recent attention (Scellier & Bengio, 2016; Lillicrap et al., 2016; Lee et al., 2015; Lansdell & Kording, 2018; Ororbia et al., 2018).

These efforts do show some ways forward. Synthetic gradients demonstrate that learning can be based on approximate gradients, and need not be temporally locked (Jaderberg et al., 2016; Czarnecki et al., 2017b). In small feedforward networks, somewhat surprisingly, fixed random feedback matrices in fact suffice for learning (Lillicrap et al., 2016) (a phenomenon known as feedback alignment). But still issues remain: feedback alignment does not work in CNNs, very deep networks, or networks with tight bottleneck layers. Regardless, these results show that rough approximations of a gradient signal can be used to learn; even relatively inefficient methods of approximating the gradient may be good enough.

On this basis, here we propose an RL algorithm to train a feedback system to enable learning. Recent work has explored similar ideas, but not with the explicit goal of approximating backpropagation (Miconi, 2017; Miconi et al., 2018; Song et al., 2017). RL-based methods like REINFORCE may be inefficient when used as a base learner, but they may be sufficient when used to train a system that itself instructs a base learner. We propose to use REINFORCE-style perturbation approach to train feedback signals to approximate what would have been provided by backpropagation.

This sort of two-learner system, where one network helps the other learn more efficiently, may in fact align well with cortical neuron physiology. For instance, the dendritic trees of pyramidal neurons consist of an apical and basal component. Such a setup has been shown to support supervised learning in feedforward networks (Guergiuev et al., 2017; Kording & Konig, 2001). Similarly, climbing fibers and Purkinje cells may define a learner/teacher system in the cerebellum (Marr, 1969). These components allow for independent integration of two different signals, and may thus provide a realistic solution to the credit assignment problem.

Thus we implement a network that learns to use feedback signals trained with reinforcement learning via a global reward signal. We mathematically analyze the model, and compare its capabilities to other methods for learning in ANNs. We prove consistency of the estimator in particular cases, extending the theory of synthetic gradient-like approaches (Jaderberg et al., 2016; Czarnecki et al., 2017b; Werbos, 1992; Schmidhuber, 1990). We demonstrate that our model learns as well as regular backpropagation in small models, overcomes the limitations of feedback alignment on more complicated feedforward networks, and can be used in convolutional networks. Thus, by combining local and global feedback signals, this method points to more plausible ways the brain could solve the credit assignment problem.

## 2  LEARNING FEEDBACK WEIGHTS THROUGH PERTURBATIONS

We use the following notation. Let $\mathbf{x} \in \mathbb{R}^m$ represent an input vector. Let an $N$ hidden-layer network be given by $\hat{\mathbf{y}} = f(\mathbf{x}) \in \mathbb{R}^p$. This is composed of a set of layer-wise summation and non-linear activations

$$\mathbf{h}^i = f^i(\mathbf{h}^{i-1}) = \sigma\left(W^i\mathbf{h}^{i-1}\right),$$

for hidden layer states $\mathbf{h}^i \in \mathbb{R}^{n_i}$, non-linearity $\sigma$, weight matrices $W^i \in \mathbb{R}^{n_i \times n_{i-1}}$ and denoting $\mathbf{h}^0 = \mathbf{x}$ and $\mathbf{h}^{N+1} = \hat{\mathbf{y}}$. Some loss function $L$ is defined in terms of the network output: $L(\mathbf{y}, \hat{\mathbf{y}})$. Let $\mathcal{L}$ denote the loss as a function of $(\mathbf{x}, \mathbf{y})$: $\mathcal{L}(\mathbf{x}, \mathbf{y}) = L(\mathbf{y}, f(\mathbf{x}))$. Let data $(\mathbf{x}, \mathbf{y}) \in \mathcal{D}$ be drawn from a distribution $\rho$. We aim to minimize: $\mathbb{E}_\rho\left[\mathcal{L}(\mathbf{x}, \mathbf{y})\right]$.

Backpropagation relies on the error signal $\mathbf{e}^i$, computed in a top-down fashion:

$$\mathbf{e}^i = \begin{cases} \partial\mathcal{L}/\partial\hat{\mathbf{y}} \circ \sigma'(W^i\mathbf{h}^{i-1}), & i = N+1; \\ \left((W^{i+1})^\mathsf{T}\mathbf{e}^{i+1}\right) \circ \sigma'(W^i\mathbf{h}^{i-1}), & 1 \le i \le N \end{cases},$$

where $\circ$ denotes element-wise multiplication.

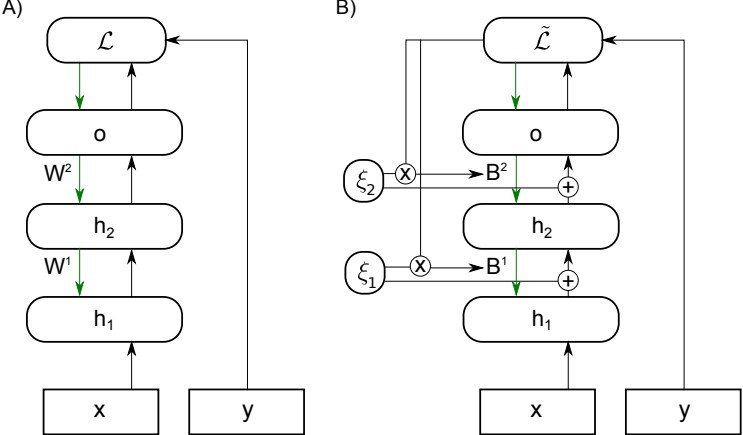

Figure 1: Learning feedback weights through perturbations. (A) Backpropagation sends error information from an output loss function, $\mathcal{L}$, through each layer from top to bottom via the same matrices $W^i$ used in the feedforward network. (B) Node perturbation introduces noise in each layer, $\xi_i$, that perturbs that layer's output and resulting loss function. The perturbed loss function, $\tilde{\mathcal{L}}$, is correlated with the noise to give an estimate of the error current. This estimate is used to update feedback matrices $B^i$ to better approximate the error signal.

## 2.1 BASIC SETUP

Let the loss gradient term be denoted as

$$\lambda^i = \frac{\partial \mathcal{L}}{\partial \mathbf{h}^i} = (W^{i+1})^\mathsf{T} \mathbf{e}^{i+1}.$$

In this work we replace $\lambda^i$ with an approximation with its own parameters to be learned (known as a synthetic gradient, or conspiring network, (Jaderberg et al., 2016; Czarnecki et al., 2017b), or error critic (Werbos, 1992)):

$$\lambda^i \approx \mathbf{g}(\mathbf{h}^i, \tilde{\mathbf{e}}^{i+1}; \theta),$$

for parameters $\theta$. Note that we must distinguish the true loss gradients from their synthetic estimates. Let $\tilde{\mathbf{e}}^i$ be loss gradients computed by backpropagating the synthetic gradients

$$\tilde{\mathbf{e}}^i = \begin{cases} \partial \mathcal{L}/\partial \hat{\mathbf{y}} \circ \sigma'(W^i \mathbf{h}^{i-1}), & i = N+1; \\ \mathbf{g}(\mathbf{h}^i, \tilde{\mathbf{e}}^{i+1}; \theta) \circ \sigma'(W^i \mathbf{h}^{i-1}), & 1 \le i \le N \end{cases}.$$

For the final layer the synthetic gradient matches the true gradient: $\mathbf{e}^{N+1} = \tilde{\mathbf{e}}^{N+1}$. This setup can accommodate both top-down and bottom-up information, and encompasses a number of published models (Jaderberg et al., 2016; Czarnecki et al., 2017b; Lillicrap et al., 2016; Nøkland, 2016; Liao et al., 2016; Xiao et al., 2018).

## 2.2 STOCHASTIC NETWORKS AND GRADIENT DESCENT

To learn a synthetic gradient we utilze the stochasticity inherent to biological neural networks. A number of biologically plausible learning rules exploit random perturbations in neural activity (Xie & Seung, 2004; Seung, 2003; Fiete & Seung, 2006; Fiete et al., 2007; Song et al., 2017). Here, at each time each unit produces a noisy response:

$$\mathbf{h}_t^i = \sigma \left( \sum_k W_{\cdot k}^i \mathbf{h}_t^{i-1} \right) + c_h \xi_t^i,$$

for independent Gaussian noise $\xi^i \sim \nu = \mathcal{N}(0, I)$ and standard deviation $c_h > 0$. This generates a noisy loss $\tilde{\mathcal{L}}(\mathbf{x}, \mathbf{y}, \xi)$ and a baseline loss $\mathcal{L}(\mathbf{x}, \mathbf{y}) = \tilde{\mathcal{L}}(\mathbf{x}, \mathbf{y}, 0)$. We will use the noisy response to estimate gradients that then allow us to optimize the baseline $\mathcal{L}$ – the gradients used for weight updates are computed using the deterministic baseline.

## 2.3 SYNTHETIC GRADIENTS VIA PERTURBATION

For Gaussian white noise, the well-known REINFORCE algorithm (Williams, 1992) coincides with the node perturbation method (Fiete & Seung, 2006; Fiete et al., 2007). Node perturbation works by linearizing the loss:

$$\tilde{\mathcal{L}} \approx \mathcal{L} + \frac{\partial \mathcal{L}}{\partial h_j^i} c_h \xi_j^i, \tag{1}$$

such that

$$\mathbb{E}\left( (\tilde{\mathcal{L}} - \mathcal{L}) c_h \xi_j^i | \mathbf{x}, \mathbf{y} \right) \approx c_h^2 \frac{\partial \mathcal{L}}{\partial h_j^i} \Big|_{\mathbf{x}, \mathbf{y}},$$

with expectation taken over the noise distribution $\nu(\xi)$. This provides an estimator of the loss gradient

$$\hat{\lambda}^i := (\tilde{\mathcal{L}}(\mathbf{x}, \mathbf{y}, \xi) - \mathcal{L}(\mathbf{x}, \mathbf{y})) \frac{\xi^i}{c_h}. \tag{2}$$

This approximation is made more precise in Theorem 1 (Supplementary material).

## 2.4 TRAINING A FEEDBACK NETWORK

There are many possible sensible choices of $\mathbf{g}(\cdot)$. For example, taking $\mathbf{g}$ as simply a function of each layer's activations: $\lambda^i = \mathbf{g}(\mathbf{h}^i)$ is in fact sufficient parameterization to express the true gradient function (Jaderberg et al., 2016). We may expect, however, that the gradient estimation problem be simpler if each layer is provided with some error information obtained from the loss function and propagated in a top-down fashion. Symmetric feedback weights may not be biologically plausible, and random fixed weights may only solve certain problems of limited size or complexity (Lillicrap et al., 2016). However, a system that can learn to appropriate feedback weights $B$ may be able to align the feedforward and feedback weights as much as is needed to successfully learn.

We investigate various choices of $\mathbf{g}(\mathbf{h}^i, \tilde{\mathbf{e}}^{i+1}; B^{i+1})$ outlined in the applications below. Parameters $B^{i+1}$ are estimated by solving the least squares problem:

$$\hat{B}^{i+1} = \arg\min_B \mathbb{E} \left\| \mathbf{g}(\mathbf{h}^i, \tilde{\mathbf{e}}^{i+1}; B) - \hat{\lambda}^i \right\|_2^2. \tag{3}$$

Unless otherwise noted this was solved by gradient-descent, updating parameters once with each minibatch. Refer to the supplementary material for additional experimental descriptions and parameters.

## 3 THEORETICAL RESULTS

We can prove the estimator (3) is consistent as the noise variance $c_h \to 0$, in some particular cases. We state the results informally here, and give the exact details in the supplementary materials. Consider first convergence of the final layer feedback matrix, $B^{N+1}$.

**Theorem 1.** *(Informal) For* $\mathbf{g}_{FA}(\mathbf{h}^i, \tilde{\mathbf{e}}^{i+1}; B^{i+1}) = B^{i+1} \tilde{\mathbf{e}}^{i+1}$, *then the least squares estimator*

$$(\hat{B}^{N+1})^{\mathsf{T}} := \hat{\lambda}^N (\mathbf{e}^{N+1})^{\mathsf{T}} \left( \mathbf{e}^{N+1} (\mathbf{e}^{N+1})^{\mathsf{T}} \right)^{-1}, \tag{4}$$

*solves (3) and converges to the true feedback matrix, in the sense that:* $\lim_{c_h \to 0} \operatorname{plim}_{T \to \infty} \hat{B}^{N+1} = W^{N+1}$, *where* plim *indicates convergence in probability.*

Theorem 1 thus establishes convergence of $B$ in a shallow (1 hidden layer) non-linear network. In a deep, linear network we can also use Theorem 1 to establish convergence over the rest of the layers.

**Theorem 2.** *(Informal) For* $\mathbf{g}_{FA}(\mathbf{h}^i, \tilde{\mathbf{e}}^{i+1}; B^{i+1}) = B^{i+1} \tilde{\mathbf{e}}^{i+1}$ *and* $\sigma(x) = x$, *the least squares estimator*

$$(\hat{B}^i)^{\mathsf{T}} := \hat{\lambda}^{i-1} (\tilde{\mathbf{e}}^i)^{\mathsf{T}} \left( \tilde{\mathbf{e}}^i (\tilde{\mathbf{e}}^i)^{\mathsf{T}} \right)^{-1} \qquad 1 \le i \le N+1, \tag{5}$$

*solves (3) and converges to the true feedback matrix, in the sense that:* $\lim_{c_h \to 0} \operatorname{plim}_{T \to \infty} \hat{B}^i = W^i$, $1 \le i \le N+1$.

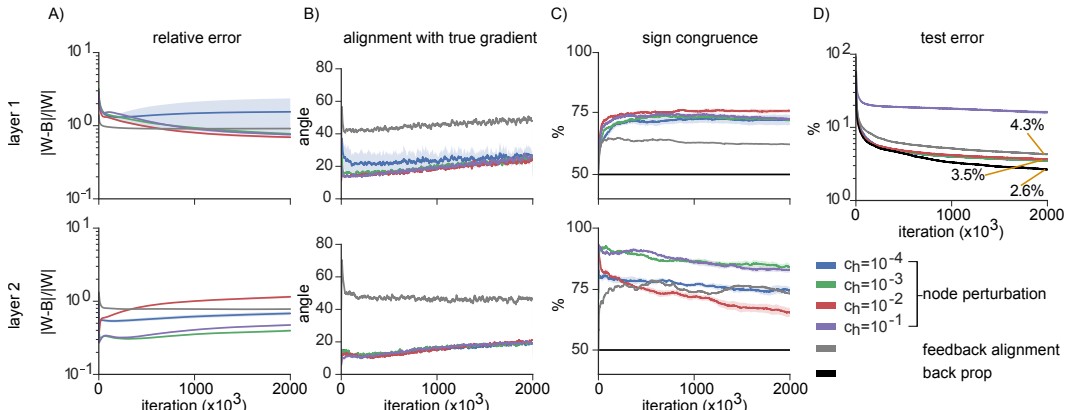

Figure 2: Node perturbation in small 4-layer network (784-50-20-10 neurons), for varying noise levels $c$, compared to feedback alignment and backpropagation. (A) Relative error between feedforward and feedback matrix. (B) Angle between true gradient and synthetic gradient estimate for each layer. (C) Percentage of signs in $W^i$ and $B^i$ that are in agreement. (D) Test error for node perturbation, backpropagation and feedback alignment. Curves show mean plus/minus standard error over 5 runs.

Given these results we can establish consistency for the 'direct feedback alignment' (DFA; Nøkland (2016)) estimator: $\mathbf{g}_{DFA}(\mathbf{h}^i, \tilde{\mathbf{e}}^{N+1}; B^{i+1}) = (B^{i+1})^{\mathsf{T}} \tilde{\mathbf{e}}^{N+1}$. Theorem 1 applies trivially since for the final layer, the two approximations have the same form: $\mathbf{g}_{FA}(\mathbf{h}^N, \tilde{\mathbf{e}}^{N+1}; \theta_N) = \mathbf{g}_{DFA}(\mathbf{h}^N, \tilde{\mathbf{e}}^{N+1}; \theta_N)$. Theorem 2 can be easily extended according to the following:

**Corollary 1.** *(Informal) For* $\mathbf{g}_{DFA}(\mathbf{h}^i, \tilde{\mathbf{e}}^{N+1}; B^{i+1}) = B^{i+1}\tilde{\mathbf{e}}^{N+1}$ *and* $\sigma(x) = x$, *the least squares estimator*

$$(\hat{B}^i)^{\mathsf{T}} := \hat{\lambda}^{i-1}(\tilde{\mathbf{e}}^{N+1})^{\mathsf{T}} \left(\tilde{\mathbf{e}}^{N+1}(\tilde{\mathbf{e}}^{N+1})^{\mathsf{T}}\right)^{-1} \qquad 1 \leq n \leq N+1, \qquad (6)$$

*solves (3) and converges to the true feedback matrix, in the sense that:* $\lim_{c_h \to 0} \text{plim}_{T \to \infty} \hat{B}^i = \prod_{j=N+1}^i W^j$, $\qquad 1 \leq i \leq N+1$.

Thus for a non-linear shallow network or a deep linear network, for both $g_{FA}$ and $g_{DFA}$, we have the result that, for sufficiently small $c_h$, if we fix the network weights $W$ and train $B$ through node perturbation then we converge to $W$. Validation that the method learns to approximate $W$, for fixed $W$, is provided in the supplementary material. In practice, we update $B$ and $W$ simultaneously. Some convergence theory is established for this case in (Jaderberg et al., 2016; Czarnecki et al., 2017b).

## 4 APPLICATIONS

### 4.1 FULLY CONNECTED NETWORKS SOLVING MNIST

First we investigate $\mathbf{g}(\mathbf{h}^i, \tilde{\mathbf{e}}^{i+1}; B^{i+1}) = (B^{i+1})^{\mathsf{T}} \tilde{\mathbf{e}}^{i+1}$, which describes a non-symmetric feedback network (Figure 1). To demonstrate the method can be used to solve simple supervised learning problems we use node perturbation with a four-layer network and MSE loss to solve MNIST (Figure 2). Updates to $W^i$ are made using the synthetic gradients $\Delta W^i = \eta \tilde{\mathbf{e}}^i \mathbf{h}^{i-1}$, for learning rate $\eta$. The feedback network needs to co-adapt with the feedforward network in order to continue to provide a useful error signal. We observed that the system is able to adjust to provide a close correspondence between the feedforward and feedback matrices in both layers of the network (Figure 2A). The relative error between $B^i$ and $W^i$ is lower than what is observed for feedback alignment, suggesting that this co-adaptation of both $W^i$ and $B^i$ is indeed beneficial. The relative error depends on the amount of noise used in node perturbation – lower variance doesn't necessarily imply the lowest

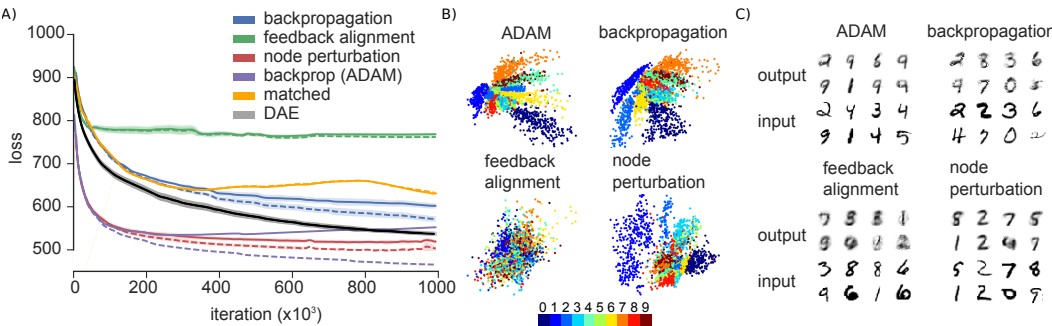

Figure 3: Results with five-layer MNIST autoencoder network. (A) Mean loss plus/minus standard error over 10 runs. Dashed lines represent training loss, solid lines represent test loss. (B) Latent space activations, colored by input label for each method. (C) Sample outputs for each method.

error between $W$ and $B$, suggesting there is an optimal noise level that balances bias in the estimate and the ability to co-adapt to the changing feedforward weights.[1]

Consistent with the low relative error in both layers, we observe that the alignment (the angle between the estimated gradient and the true gradient – proportional to $\mathbf{e}^\mathsf{T} W B^\mathsf{T} \tilde{\mathbf{e}}$) is low in each layer – much lower for node perturbation than for feedback alignment, again suggesting that the method is much better at communicating error signals between layers (Figure 2B). In fact, recent studies have shown that sign congruence of the feedforward and feedback matrices is all that is required to achieve good performance (Liao et al., 2016; Xiao et al., 2018). Here the sign congruence is also higher in node perturbation, again depending somewhat the variance. The amount of congruence is comparable between layers (Figure 2C). Finally, the learning performance of node perturbation is comparable to backpropagation (Figure 2D), and better than feedback alignment in this case, though not by much. Note that by setting the feedback learning rate to zero, we recover the feedback alignment algorithm. So we should expect to be always able to do at least as well as feedback alignment. These results instead highlight the qualitative differences between the methods, and suggest that node perturbation for learning feedback weights can be used to approximate gradients in deep networks.

## 4.2 Auto-encoding MNIST

The above results demonstrate node perturbation provides error signals closely aligned with the true gradients. However, performance-wise they do not demonstrate any clear advantage over feedback alignment or backpropagation. A known shortcoming of feedback alignment is in very deep networks and in autoencoding networks with tight bottleneck layers (Lillicrap et al., 2016). To see if node perturbation has the same shortcoming, we test performance of a $\mathbf{g}(\mathbf{h}^i, \tilde{\mathbf{e}}^{i+1}; B^{i+1}) = (B^{i+1})^\mathsf{T} \tilde{\mathbf{e}}^{i+1}$ model on a simple auto-encoding network with MNIST input data (size 784-200-2-200-784). In this more challenging case we also compare the method to the 'matching' learning rule (Rombouts et al., 2015; Martinolli et al., 2018), in which updates to $B$ match updates to $W$ and weight decay is added, a denoising autoencoder (DAE) (Vincent et al., 2008), and the ADAM (Kingma & Ba, 2015) optimizer (with backprop gradients).

As expected, feedback alignment performs poorly, while node perturbation performs better than backpropagation (Figure 3A). The increased performance relative to backpropagation may seem surprising. A possible reason is the addition of noise in our method encourages learning of more robust latent factors (Alain & Bengio, 2015). The DAE also improves the loss over vanilla backpropagation (Figure 3A). And, in line with these ideas, the latent space learnt by node perturbation shows a more uniform separation between the digits, compared to the networks trained by backpropagation. Feedback alignment, in contrast, does not learn to separate digits in the bottleneck layer at all (Figure 3B), resulting in scrambled output (Figure 3C). The matched learning rule performs similarly to backpropagation. These possible explanations are investigated more below. Regardless,

---

[1]Code to reproduce these results can be found at: `https://github.com/benlansdell/synthfeedback`

these results show that node perturbation is able to successfully communicate error signals through thin layers of a network as needed.

## 4.3 CONVOLUTIONAL NEURAL NETWORKS SOLVING CIFAR

Convolutional networks are another known shortcoming of feedback alignment. Here we test the method on a convolutional neural network (CNN) solving CIFAR (Krizhevsky, 2009). Refer to the supplementary material for architecture and parameter details. For this network we learn feedback weights direct from the output layer to each earlier layer: $\mathbf{g}(\mathbf{h}^i, \tilde{\mathbf{e}}^{i+1}; B^{i+1}) = (B^{i+1})^\mathsf{T} \tilde{\mathbf{e}}^{N+1}$ (similar to 'direct feedback alignment' (Nøkland, 2016)). Here this was solved by gradient-descent. On CIFAR10 we obtain a test accuracy of 75%. When compared with fixed feedback weights and backpropagation, we see it is advantageous to learn feedback weights on CIFAR10 and marginally advantageous on CIFAR100 (Table 1). This shows the method can be used in a CNN, and can solve challenging computer vision problems without weight transport.

Table 1: Mean test accuracy of CNN over 5 runs trained with backpropagation, node perturbation and direct feedback alignment (DFA) (Nøkland, 2016; Crafton et al., 2019).

| dataset | backpropagation | node perturbation | DFA |
|---------|-----------------|-------------------|-----|
| CIFAR10 | 76.9±0.1 | 74.8±0.2 | 72.4±0.2 |
| CIFAR100 | 51.2±0.1 | 48.1±0.2 | 47.3±0.1 |

## 4.4 WHAT IS HELPING, NOISY ACTIVATIONS OR APPROXIMATING THE GRADIENT?

To solve the credit assignment problem, our method utilizes two well-explored strategies in deep learning: adding noise (generally used to regularize (Bengio et al., 2013; Gulcehre et al., 2016; Neelakantan et al., 2015; Bishop, 1995)), and approximating the true gradients (Jaderberg et al., 2016). To determine which of these features are responsible for the improvement in performance over fixed weights, in the autoencoding and CIFAR10 cases, we study the performance while varying where noise is added to the models (Table 2). Noise can be added to the activations (BP and FA w. noise, Table 2), or to the inputs, as in a denoising autoencoder (DAE, Table 2). Or, noise can be used only in obtaining an estimator of the true gradients (as in our method; NP, Table 2). For comparison, a noiseless version of our method must instead assume access to the true gradients, and use this to learn feedback weights (i.e. synthetic gradients (Jaderberg et al., 2016); SG, Table 2). Each of these models is tested on the autoencoding and CIFAR10 tasks, allowing us to better understand the performance of the node perturbation method.

Table 2: Mean loss (plus/minus standard error) on autoencoding MNIST task (left) and mean accuracy on CIFAR10 task (right). Shaded cells indicate methods which do not use weight transport or exact gradient supervision. Best performance indicated in **boldface**. Implementation details of each method is provided in the supplementary material.

(a) MNIST autoencoder

| method | noise | no noise |
|--------|-------|----------|
| BP(SGD) | 536.8±2.1 | 609.8±14.4 |
| BP(ADAM) | 522.3±0.4 | 533.3±2.2 |
| FA | 768.2±2.7 | 759.1±3.3 |
| DAE | 539.8±4.9 | – |
| NP (ours) | **515.3±4.1** | – |
| SG | – | 521.6±2.3 |
| Matched | 629.9±1.1 | 615.0±0.4 |

(b) CIFAR10 classification

| method | noise | no noise |
|--------|-------|----------|
| BP | 76.8±0.2 | **76.9±0.1** |
| DFA | 72.4±0.2 | 72.3±0.1 |
| NP (ours) | 74.8±0.2 | – |
| SG | – | 75.3±0.3 |

In the autoencoding task, both noise (either in the inputs or the activations) and using an approximator to the gradient improve performance (Table 2, left). Noise benefits performance for both SGD optimization and ADAM (Kingma & Ba, 2015). In fact in this task, the combination of both of

these factors (i.e. our method) results in better performance over either alone. Yet, the addition of noise to the activations does not help feedback alignment. This suggests that our method is indeed learning useful approximations of the error signals, and is not merely improving due to the addition of noise to the system. In the CIFAR10 task (Table 2, right), the addition of noise to the activations has minimal effect on performance, while having access to the true gradients (SG) does result in improved performance over fixed feedback weights. Thus in these tasks it appears that noise does not always help, but using a less-based gradient estimator does, and noisy activations are one way of obtaining an unbiased gradient estimator. Our method also is the best performing method that does not require either weight transport or access to the true gradients as a supervisory signal.

## 5 DISCUSSION

Here we implement a perturbation-based synthetic gradient method to train neural networks. We show that this hybrid approach can be used in both fully connected and convolutional networks. By removing the symmetric feedforward/feedback weight requirement imposed by backpropagation, this approach is a step towards more biologically-plausible deep learning. By reaching comparable performance to backpropagation on MNIST, the method is able to solve larger problems than perturbation-only methods (Xie & Seung, 2004; Fiete et al., 2007; Werfel et al., 2005). By working in cases that feedback alignment fails, the method can provide learning without weight transport in a more diverse set of network architectures. We thus believe the idea of integrating both local and global feedback signals is a promising direction towards biologically plausible learning algorithms.

Of course, the method does not solve all issues with implementing gradient-based learning in a biologically plausible manner. For instance, in the current implementation, the forward and the backwards passes are locked. Here we just focus on the weight transport problem. A current drawback is that the method does not reach state-of-the-art performance on more challenging datasets like CIFAR. We focused on demonstrating that it is advantageous to learn feedback weights, when compared with fixed weights, and successfully did so in a number of cases. However, we did not use any additional data augmentation and regularization methods often employed to reach state-of-the-art performance. Thus fully characterizing the performance of this method remains important future work. The method also does not tackle the temporal credit assignment problem, which has also seen recent progress in biologically plausible implementation Ororbia et al. (2019b;a).

However the method does has a number of computational advantages. First, without weight transport the method has better data-movement performance (Crafton et al., 2019; Akrout et al., 2019), meaning it may be more efficiently implemented than backpropagation on specialized hardware. Second, by relying on random perturbations to measure gradients, the method does not rely on the environment to provide gradients (compared with e.g. Czarnecki et al. (2017a); Jaderberg et al. (2016)). Our theoretical results are somewhat similar to that of Alain & Bengio (2015), who demonstrate that a denoising autoencoder converges to the unperturbed solution as Gaussian noise goes to zero. However our results apply to subgaussian noise more generally.

While previous research has provided some insight and theory for how feedback alignment works (Lillicrap et al., 2016; Ororbia et al., 2018; Moskovitz et al., 2018; Bartunov et al., 2018; Baldi et al., 2018) the effect remains somewhat mysterious, and not applicable in some network architectures. Recent studies have shown that some of these weaknesses can be addressed by instead imposing sign congruent feedforward and feedback matrices (Xiao et al., 2018). Yet what mechanism may produce congruence in biological networks is unknown. Here we show that the shortcomings of feedback alignment can be addressed in another way: the system can learn to adjust weights as needed to provide a useful error signal. Our work is closely related to Akrout et al. (2019), which also uses perturbations to learn feedback weights. However our approach does not divide learning into two phases, and training of the feedback weights does not occur in a layer-wise fashion, assuming only one layer is noisy at a time, which is a strong assumption. Here instead we focus on combining global and local learning signals.

Here we tested our method in an idealized setting. However the method is consistent with neurobiology in two important ways. First, it involves separate learning of feedforward and feedback weights. This is possible in cortical networks, where complex feedback connections exist between layers (Lacefield et al., 2019; Richards & Lillicrap, 2019) and pyramidal cells have apical and basal compartments that allow for separate integration of feedback and feedforward signals (Guerguiev

et al., 2017; Körding & König, 2001). A recent finding that apical dendrites receive reward information is particularly interesting (Lacefield et al., 2019). Models like Guerguiev et al. (2017) show how the ideas in this paper may be implemented in spiking neural networks. We believe such models can be augmented with a perturbation-based rule like ours to provide a better learning system.

The second feature is that perturbations are used to learn the feedback weights. How can a neuron measure these perturbations? There are many plausible mechanisms (Seung, 2003; Xie & Seung, 2004; Fiete & Seung, 2006; Fiete et al., 2007). For instance, birdsong learning uses empiric synapses from area LMAN (Fiete et al., 2007), others proposed it is approximated (Legenstein et al., 2010; Hoerzer et al., 2014), or neurons could use a learning rule that does not require knowing the noise (Lansdell & Kording, 2018). Further, our model involves the subtraction of a baseline loss to reduce the variance of the estimator. This does not affect the expected value of the estimator – technically the baseline could be removed or replaced with an approximation (Legenstein et al., 2010; Loewenstein & Seung, 2006). Thus both separation of feedforward and feedback systems and perturbation-based estimators can be implemented by neurons.

As RL-based methods do not scale by themselves, and exact gradient signals are infeasible, the brain may well use a feedback system trained through reinforcement signals to usefully approximate gradients. There is a large space of plausible learning rules that can learn to use feedback signals in order to more efficiently learn, and these promise to inform both models of learning in the brain and learning algorithms in artificial networks. Here we take an early step in this direction.

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

# A  PROOFS

We review the key components of the model. Data $(\mathbf{x}, \mathbf{y}) \in \mathcal{D}$ are drawn from a distribution $\rho$. The loss function is linearized:

$$\tilde{\mathcal{L}} \approx \mathcal{L} + \frac{\partial \mathcal{L}}{\partial h_j^i} c_h \xi_j^i, \tag{7}$$

such that

$$\mathbb{E}\left( (\tilde{\mathcal{L}} - \mathcal{L}) c_h \xi_j^i | \mathbf{x}, \mathbf{y} \right) \approx c_h^2 \frac{\partial \mathcal{L}}{\partial h_j^i}\bigg|_{\mathbf{x},\mathbf{y}},$$

with expectation taken over the noise distribution $\nu(\xi)$. This suggests a good estimator of the loss gradient is

$$\hat{\lambda}^i := (\tilde{\mathcal{L}}(\mathbf{x}, \mathbf{y}, \xi) - \mathcal{L}(\mathbf{x}, \mathbf{y}))\frac{\xi^i}{c_h}. \tag{8}$$

Let $\tilde{\mathbf{e}}^i$ be the error signal computed by backpropagating the synthetic gradients:

$$\tilde{\mathbf{e}}^i = \begin{cases} \partial \mathcal{L}/\partial \hat{\mathbf{y}} \circ \sigma'(W^i \mathbf{h}^{i-1}), & i = N + 1; \\ \left( (\hat{B}^{i+1})^\mathsf{T} \tilde{\mathbf{e}}^{i+1} \right) \circ \sigma'(W^i \mathbf{h}^{i-1}), & 1 \le i \le N. \end{cases}$$

Then parameters $B^{i+1}$ are estimated by solving the least squares problem:

$$\hat{B}^{i+1} = \arg\min_B \mathbb{E} \left\| B^\mathsf{T} \tilde{\mathbf{e}}^{i+1} - \hat{\lambda}^i \right\|_2^2. \tag{9}$$

Note that the matrix-vector form of backpropagation given here is setup so that we can think of each term as either a vector for a single input, or as matrices corresponding to a set of $T$ inputs. Here we focus on the question, under what conditions can we show that $\hat{B}^{i+1} \to W^{i+1}$, as $T \to \infty$?

One way to find an answer is to define the synthetic gradient in terms of the system without noise added. Then $B^\mathsf{T} \tilde{\mathbf{e}}$ is deterministic with respect to $\mathbf{x}, \mathbf{y}$ and, assuming $\tilde{\mathcal{L}}$ has a convergent power series around $\xi = 0$, we can write

$$\mathbb{E}(\hat{\lambda}^i | \mathbf{x}, \mathbf{y}) = \mathbb{E}\left( \frac{1}{c_h^2} \left[ \frac{\partial \mathcal{L}}{\partial h^i} (c_h \xi_j^i)^2 + \sum_{m=2}^{\infty} \frac{\mathcal{L}_{ij}^{(m)}}{m!} (c_h \xi_j^i)^{m+1} \right] \bigg| \mathbf{x}, \mathbf{y} \right)$$

$$= (W^{i+1})^\mathsf{T} \mathbf{e}^{i+1} + \mathbb{E}\left( \frac{1}{c_h^2} \sum_{m=2}^{\infty} \frac{\mathcal{L}_{ij}^{(m)}}{m!} (c_h \xi_j^i)^{m+1} | \mathbf{x}, \mathbf{y} \right).$$

Taken together these suggest we can prove $\hat{B}^{i+1} \to W^{i+1}$ in the same way we prove consistency of the linear least squares estimator.

For this to work we must show the expectation of the Taylor series approximation (1) is well behaved. That is, we must show the expected remainder term of the expansion:

$$\mathcal{E}_j^i(c_h) = \mathbb{E}\left[ \frac{1}{c_h^2} \sum_{m=2}^{\infty} \frac{\mathcal{L}_{ij}^{(m)}}{m!} (c_h \xi_j^i)^{m+1} | \mathbf{x}, \mathbf{y} \right],$$

is finite and goes to zero as $c_h \to 0$. This requires some additional assumptions on the problem.

We make the following assumptions:

- A1: the noise $\xi$ is subgaussian,
- A2: the loss function $\mathcal{L}(\mathbf{x}, \mathbf{y})$ is analytic on $\mathcal{D}$,
- A3: the error matrices $\tilde{\mathbf{e}}^i(\tilde{\mathbf{e}}^i)^\mathsf{T}$ are full rank, for $1 \le i \le N + 1$, with probability 1,
- A4: the mean of the remainder and error terms is bounded:

$$\mathbb{E}\left[ \mathcal{E}^i(c_h)(\tilde{\mathbf{e}}^{i+1})^\mathsf{T} \right] < \infty,$$

for $1 \le i \le N$.

Consider first convergence of the final layer feedback matrix, $B^{N+1}$. In the final layer it is true that $\mathbf{e}^{N+1} = \tilde{\mathbf{e}}^{N+1}$.

**Theorem 1.** *Assume A1-4. For* $\mathbf{g}_{FA}(\mathbf{h}^i, \tilde{\mathbf{e}}^{i+1}; B^{i+1}) = B^{i+1}\tilde{\mathbf{e}}^{i+1}$, *then the least squares estimator*

$$(\hat{B}^{N+1})^\mathsf{T} := \hat{\lambda}^N (\mathbf{e}^{N+1})^\mathsf{T} \left( \mathbf{e}^{N+1}(\mathbf{e}^{N+1})^\mathsf{T} \right)^{-1}, \qquad (10)$$

*solves (3) and converges to the true feedback matrix, in the sense that:*

$$\lim_{c_h \to 0} \underset{T \to \infty}{\mathrm{plim}} \; \hat{B}^{N+1} = W^{N+1}.$$

*Proof.* Let $\mathcal{L}_{ij}^{(m)} := \frac{\partial^m \mathcal{L}}{\partial h_j^{im}}$. We first show that, under A1-2, the conditional expectation of the estimator (2) converges to the gradient $\mathcal{L}_{Nj}^{(1)}$ as $c_h \to 0$. For each $\hat{\lambda}_j^N$, by A2, we have the following series expanded around $\xi = 0$:

$$\hat{\lambda}_j^N = \frac{1}{c_h^2} \sum_{m=1}^{\infty} \frac{\mathcal{L}_{ij}^{(m)}}{m!} (c_h \xi_j^N)^{m+1}.$$

Taking a conditional expectation gives:

$$\mathbb{E}(\hat{\lambda}_j^N | \mathbf{x}, \mathbf{y}) = (W^{N+1})^\mathsf{T} \mathbf{e}^{N+1} + \mathbb{E}\left[ \frac{1}{c_h^2} \sum_{m=2}^{\infty} \frac{\mathcal{L}_{Nj}^{(m)}}{m!} (c_h \xi_j^N)^{m+1} | \mathbf{x}, \mathbf{y} \right].$$

We must show the remainder term

$$\mathcal{E}^N(c_h) = \mathbb{E}\left[ \frac{1}{c_h^2} \sum_{m=2}^{\infty} \frac{\mathcal{L}_{Nj}^{(m)}}{m!} (c_h \xi_j^N)^{m+1} | \mathbf{x}, \mathbf{y} \right],$$

goes to zero as $c_h \to 0$. This is true provided each moment $\mathbb{E}((\xi_j^N)^m | \mathbf{x}, \mathbf{y})$ is sufficiently well-behaved. Using Jensen's inequality and the triangle inequality in the first line, we have that

$$|\mathcal{E}^N(c_h)| \leq \mathbb{E}\left[ \frac{1}{c_h^2} \sum_{m=2}^{\infty} \left| \frac{\mathcal{L}_{Nj}^{(m)}}{m!} \right| |c_h \xi_j^N|^{m+1} | \mathbf{x}, \mathbf{y} \right], \quad \forall (\mathbf{x}, \mathbf{y}) \in \mathcal{D}$$

$$[\text{monotone convergence}] \quad = \sum_{m=2}^{\infty} \left| \frac{\mathcal{L}_{Nj}^{(m)}}{m!} \right| (c_h)^{m-1} \mathbb{E}\left[ |\xi_j^N|^{m+1} \right]$$

$$[\text{subgaussian}] \quad \leq K \sum_{m=2}^{\infty} \left| \frac{\mathcal{L}_{Nj}^{(m)}}{m!} \right| (c_h)^{m-1} (\sqrt{m+1})^{m+1}$$

$$= \mathcal{O}(c_h) \qquad \text{as } c_h \to 0. \qquad (11)$$

With this in place, we have that the problem (9) is close to a linear least squares problem, since

$$\hat{\lambda}^N = (W^{N+1})^\mathsf{T} \mathbf{e}^{N+1} + \mathcal{E}^N(c_h) + \eta^N, \qquad (12)$$

with residual $\eta^N = \hat{\lambda}^N - \mathbb{E}(\hat{\lambda}^N | \mathbf{x}, \mathbf{y})$. The residual satisfies

$$\mathbb{E}\left( \mathbf{e}^{N+1}(\eta^N)^\mathsf{T} \right) = \mathbb{E}(\mathbf{e}^{N+1}(\hat{\lambda}^N)^\mathsf{T} - \mathbf{e}^{N+1}\mathbb{E}((\hat{\lambda}^N)^\mathsf{T} | \mathbf{x}, \mathbf{y}))$$

$$= \mathbb{E}\left( \mathbf{e}^{N+1}(\hat{\lambda}^N)^\mathsf{T} - \mathbb{E}\left( \mathbf{e}^{N+1}(\hat{\lambda}^N)^\mathsf{T} | \mathbf{x}, \mathbf{y} \right) \right)$$

$$= 0. \qquad (13)$$

This follows since $\mathbf{e}^{N+1}$ is defined in relation to the baseline loss, not the stochastic loss, meaning it is measurable with respect to $(\mathbf{x}, \mathbf{y})$ and can be moved into the conditional expectation.

From (12) and A3, we have that the least squares estimator (10) satisfies

$$(\hat{B}^{N+1})^\mathsf{T} = (W^{N+1})^\mathsf{T} + (\mathcal{E}^N(c_h) + \eta^N)(\mathbf{e}^{N+1})^\mathsf{T}(\mathbf{e}^{N+1}(\mathbf{e}^{N+1})^\mathsf{T})^{-1}.$$

Thus, using the continuous mapping theorem

$$
\begin{aligned}
\operatorname*{plim}_{T\to\infty}(\hat{B}^{N+1})^{\mathsf{T}} &= (W^{N+1})^{\mathsf{T}} + \left[\operatorname*{plim}_{T\to\infty}\frac{1}{T}(\mathcal{E}^N(c_h)+\eta^N)(\mathbf{e}^{N+1})^{\mathsf{T}}\right]\left[\operatorname*{plim}_{T\to\infty}\frac{1}{T}\mathbf{e}^{N+1}(\mathbf{e}^{N+1})^{\mathsf{T}}\right]^{-1}\\
[\text{WLLN}] \quad &= (W^{N+1})^{\mathsf{T}} + \mathbb{E}\left[(\mathcal{E}(c_h)+\eta^N)(\mathbf{e}^{N+1})^{\mathsf{T}}\right]\left[\mathbb{E}(\mathbf{e}^{N+1}(\mathbf{e}^{N+1})^{\mathsf{T}})\right]^{-1}\\
[\text{Eq. (13)}] \quad &= (W^{N+1})^{\mathsf{T}} + \mathbb{E}\left[\mathcal{E}(c_h)(\mathbf{e}^{N+1})^{\mathsf{T}}\right]\left[\mathbb{E}(\mathbf{e}^{N+1}(\mathbf{e}^{N+1})^{\mathsf{T}})\right]^{-1}\\
[\text{A4 and Eq. (11)}] \quad &= (W^{N+1})^{\mathsf{T}} + \mathcal{O}(c_h).
\end{aligned}
$$

Then we have:
$$
\lim_{c_h\to 0}\operatorname*{plim}_{T\to\infty}\hat{B}^{N+1} = W^{N+1}.
$$

□

We can use Theorem 1 to establish convergence over the rest of the layers of the network when the activation function is the identity.

**Theorem 2.** *Assume A1-4. For* $\mathbf{g}_{FA}(\mathbf{h}^i, \tilde{\mathbf{e}}^{i+1}; B^{i+1}) = B^{i+1}\tilde{\mathbf{e}}^{i+1}$ *and* $\sigma(x) = x$, *the least squares estimator*

$$
(\hat{B}^i)^{\mathsf{T}} := \hat{\lambda}^{i-1}(\tilde{\mathbf{e}}^i)^{\mathsf{T}}\left(\tilde{\mathbf{e}}^i(\tilde{\mathbf{e}}^i)^{\mathsf{T}}\right)^{-1} \qquad 1 \le i \le N+1, \tag{14}
$$

*solves (9) and converges to the true feedback matrix, in the sense that:*

$$
\lim_{c_h\to 0}\operatorname*{plim}_{T\to\infty}\hat{B}^i = W^i, \qquad 1 \le i \le N+1.
$$

*Proof.* Define
$$
\tilde{W}^i(c) := \operatorname*{plim}_{T\to\infty}\hat{B}^i,
$$

assuming this limit exists. From Theorem 1 the top layer estimate $\hat{B}^{N+1}$ converges in probability to $\tilde{W}^{N+1}(c)$.

We can then use induction to establish that $\hat{B}^j$ in the remaining layers also converges in probability to $\tilde{W}^j(c)$. That is, assume that $\hat{B}^j$ converge in probability to $\tilde{W}^j(c)$ in higher layers $N+1 \ge j > i$. Then we must establish that $\hat{B}^i$ also converges in probability.

To proceed it is useful to also define
$$
\tilde{\mathbf{e}}(c)^i := \begin{cases} \partial\mathcal{L}/\partial\hat{\mathbf{y}} \circ \sigma'(W^i\mathbf{h}^{i-1}), & i = N+1;\\ \left((\tilde{W}^{i+1}(c))^{\mathsf{T}}\tilde{\mathbf{e}}^{i+1}\right)\circ\sigma'(W^i\mathbf{h}^{i-1}), & 1 \le i \le N, \end{cases}
$$

as the error signal backpropagated through the converged (but biased) weight matrices $\tilde{W}(c)$. Again it is true that $\tilde{\mathbf{e}}^{N+1} = \mathbf{e}^{N+1}$.

As in Theorem 1, the least squares estimator has the form:
$$
(\hat{B}^i)^{\mathsf{T}} = \hat{\lambda}^{i-1}(\tilde{\mathbf{e}}^i)^{\mathsf{T}}\left(\tilde{\mathbf{e}}^i(\tilde{\mathbf{e}}^i)^{\mathsf{T}}\right)^{-1}.
$$

Thus, again by the continuous mapping theorem:
$$
\begin{aligned}
\operatorname*{plim}_{T\to\infty}(\hat{B}^i)^{\mathsf{T}} &= \left[\operatorname*{plim}_{T\to\infty}\frac{1}{T}\hat{\lambda}^{i-1}(\tilde{\mathbf{e}}^i)^{\mathsf{T}}\right]\left[\operatorname*{plim}_{T\to\infty}\frac{1}{T}\tilde{\mathbf{e}}^i(\tilde{\mathbf{e}}^i)^{\mathsf{T}}\right]^{-1}\\
&= \left[\operatorname*{plim}_{T\to\infty}\frac{1}{T}\hat{\lambda}^{i-1}(\mathbf{e}^{N+1})^{\mathsf{T}}\hat{B}^{N+1}\cdots\hat{B}^{i+1}\right]\left[\operatorname*{plim}_{T\to\infty}\frac{1}{T}\tilde{\mathbf{e}}^i(\tilde{\mathbf{e}}^i)^{\mathsf{T}}\right]^{-1}
\end{aligned}
$$

In this case continuity again allows us to separate convergence of each term in the product:
$$
\operatorname*{plim}_{T\to\infty}\frac{1}{T}\hat{\lambda}^{i-1}(\mathbf{e}^{N+1})^{\mathsf{T}}\hat{B}^{N+1}\cdots\hat{B}^{i+1} = \left[\operatorname*{plim}_{T\to\infty}\frac{1}{T}\hat{\lambda}^{i-1}(\mathbf{e}^{N+1})^{\mathsf{T}}\right]\left[\operatorname*{plim}_{T\to\infty}\hat{B}^{N+1}\right]\cdots\left[\operatorname*{plim}_{T\to\infty}\hat{B}^{i+1}\right]
$$
$$
\tag{15}
$$
$$
\begin{aligned}
&= \mathbb{E}(\hat{\lambda}^{i-1}(\mathbf{e}^{N+1})^{\mathsf{T}})W^{N+1}(c)\cdots W^{i+1}(c),\\
&= \mathbb{E}(\hat{\lambda}^{i-1}(\tilde{\mathbf{e}}^i(c))^{\mathsf{T}})
\end{aligned}
$$

using the weak law of large numbers in the first term, and the induction assumption for the remaining terms. In the same way

$$\operatorname*{plim}_{T\to\infty} \frac{1}{T}\tilde{\mathbf{e}}^i(\tilde{\mathbf{e}}^i)^\mathsf{T} = \mathbb{E}(\tilde{\tilde{\mathbf{e}}}^i(c)(\tilde{\tilde{\mathbf{e}}}^i(c))^\mathsf{T}).$$

Note that the induction assumption also implies $\lim_{c\to 0} \tilde{\tilde{\mathbf{e}}}^i(c) = \mathbf{e}^i$. Thus, putting it together, by A3, A4 and the same reasoning as in Theorem 1 we have the result:

$$\lim_{c_h\to 0} \operatorname*{plim}_{T\to\infty} (\hat{B}^i)^\mathsf{T} = \lim_{c\to 0} \left[ (W^i)^\mathsf{T} \mathbb{E}(\mathbf{e}^i(\tilde{\tilde{\mathbf{e}}}^i(c))^\mathsf{T}) + \mathbb{E}(\mathcal{E}^{i-1}(c)(\tilde{\tilde{\mathbf{e}}}^i(c))^\mathsf{T}) \right] \left[ \mathbb{E}(\tilde{\tilde{\mathbf{e}}}^i(c)(\tilde{\tilde{\mathbf{e}}}^i(c))^\mathsf{T}) \right]^{-1}$$

$$= (W^i)^\mathsf{T}.$$

$\square$

**Corollary 1.** *Assume A1-4. For* $\mathbf{g}_{DFA}(\mathbf{h}^i, \tilde{\mathbf{e}}^{N+1}; B^{i+1}) = B^{i+1}\tilde{\mathbf{e}}^{N+1}$ *and* $\sigma(x) = x$, *the least squares estimator*

$$(\hat{B}^i)^\mathsf{T} := \hat{\lambda}^{i-1}(\tilde{\mathbf{e}}^{N+1})^\mathsf{T} \left( \tilde{\mathbf{e}}^{N+1}(\tilde{\mathbf{e}}^{N+1})^\mathsf{T} \right)^{-1} \qquad 1 \le i \le N+1, \tag{16}$$

*solves (3) and converges to the true feedback matrix, in the sense that:*

$$\lim_{c_h\to 0} \operatorname*{plim}_{T\to\infty} \hat{B}^i = \prod_{j=N+1}^{i} W^j, \qquad 1 \le i \le N+1.$$

*Proof.* For a deep linear network notice that the node perturbation estimator can be expressed as:

$$\hat{\lambda}^i = (W^{i+1}\cdots W^{N+1})^\mathsf{T}\mathbf{e}^{N+1} + \mathcal{E}^i(c_h) + \eta^i, \tag{17}$$

where the first term represents the true gradient, given by the simple linear backpropagation, the second and third terms are the remainder and a noise term, as in Theorem 1. Define

$$V^i := \prod_{j=N+1}^{i} W_j.$$

Then following the same reasoning as the proof of Theorem 1, we have:

$$\operatorname*{plim}_{T\to\infty} (\hat{B}^{i+1})^\mathsf{T} = (V^{i+1})^\mathsf{T} + \left[ \operatorname*{plim}_{T\to\infty} \frac{1}{T}(\mathcal{E}^i(c_h) + \eta^i)(\mathbf{e}^{N+1})^\mathsf{T} \right] \left[ \operatorname*{plim}_{T\to\infty} \frac{1}{T}\mathbf{e}^{N+1}(\mathbf{e}^{N+1})^\mathsf{T} \right]^{-1}$$

$$= (V^{i+1})^\mathsf{T} + \mathbb{E}\left[ (\mathcal{E}(c_h) + \eta^i)(\mathbf{e}^{N+1})^\mathsf{T} \right] \left[ \mathbb{E}(\mathbf{e}^{N+1}(\mathbf{e}^{N+1})^\mathsf{T}) \right]^{-1}$$

$$= (V^{i+1})^\mathsf{T} + \mathbb{E}\left[ \mathcal{E}(c_h)(\mathbf{e}^{N+1})^\mathsf{T} \right] \left[ \mathbb{E}(\mathbf{e}^{N+1}(\mathbf{e}^{N+1})^\mathsf{T}) \right]^{-1}$$

$$= (V^{i+1})^\mathsf{T} + \mathcal{O}(c_h).$$

Then we have:

$$\lim_{c_h\to 0} \operatorname*{plim}_{T\to\infty} \hat{B}^{i+1} = V^{i+1}.$$

$\square$

### A.1 Discussion of assumptions

It is worth making the following points on each of the assumptions:

- A1. In the paper we assume $\xi$ is Gaussian. Here we prove the more general result of convergence for any subgaussian random variable.

- A2. In practice this may be a fairly restrictive assumption, since it precludes using relu non-linearities. Other common choices, such as hyperbolic tangent and sigmoid non-linearities with an analytic cost function do satisfy this assumption, however.

- A3. It is hard to establish general conditions under which $\tilde{\mathbf{e}}^i(\tilde{\mathbf{e}}^i)^\mathsf{T}$ will be full rank. While it may be a reasonable assumption in some cases.

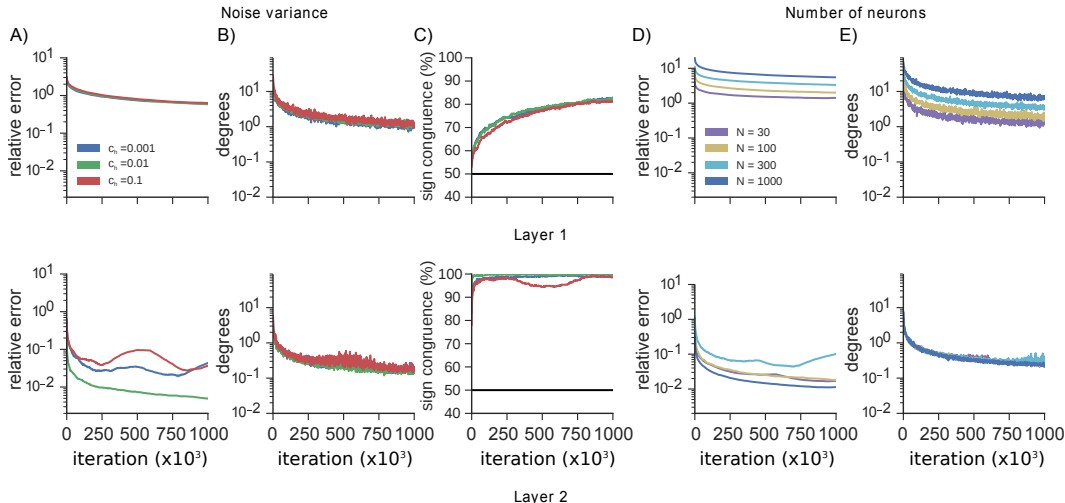

Figure 4: Convergence of node perturbation method in a two hidden layer neural network (784-50-20-10) with MSE loss, for varying noise levels $c$. Node perturbation is used to estimate feedback matrices that provide gradient estimates for fixed $W$. (A) Relative error ($\|W^i - B^i\|_F/\|W^i\|_F$) for each layer. (B) Angle between true gradient and synthetic gradient estimate at each layer. (C) Percentage of signs in $W^i$ and $B^i$ that are in agreement. (D) Relative error when number of neurons is varied (784-N-50-10). (E) Angle between true gradient and synthetic gradient estimate at each layer.

Extensions of Theorem 2 to a non-linear network may be possible. However, the method of proof used here is not immediately applicable because the continuous mapping theorem can not be applied in such a straightforward fashion as in Equation (15). In the non-linear case the resulting sums over all observations are neither independent or identically distributed, which makes applying any law of large numbers complicated.

## B  VALIDATION WITH FIXED $W$

We demonstrate the method's convergence in a small non-linear network solving MNIST for different noise levels, $c_h$, and layer widths (Figure 4). As basic validation of the method, in this experiment the feedback matrices are updated while the feedforward weights $W^i$ are held fixed. We should expect the feedback matrices $B^i$ to converge to the feedforward matrices $W^i$. Here different noise variance does results equally accurate estimators (Figure 4A). The estimator correctly estimates the true feedback matrix $W^2$ to a relative error of 0.8%. The convergence is layer dependent, with the second hidden layer matrix, $W^2$, being accurately estimated, and the convergence of the first hidden layer matrix, $W^1$, being less accurately estimated. Despite this, the angles between the estimated gradient and the true gradient (proportional to $\mathbf{e}^\mathsf{T} W B^\mathsf{T} \tilde{\mathbf{e}}$) are very close to zero for both layers (Figure 4B) (less than 90 degrees corresponds to a descent direction). Thus the estimated gradients strongly align with true gradients in both layers. Recent studies have shown that sign congruence of the feedforward and feedback matrices is all that is required to achieve good performance Liao et al. (2016); Xiao et al. (2018). Here significant sign congruence is achieved in both layers (Figure 4C), despite the matrices themselves being quite different in the first layer. The number of neurons has an effect on both the relative error in each layer and the extent of alignment between true and synthetic gradient (Figure 4D,E). The method provides useful error signals for a variety of sized networks, and can provide useful error information to layers through a deep network.

## C  EXPERIMENT DETAILS

Details of each task and parameters are provided here. All code is implemented in TensorFlow.

## C.1 FIGURE 2

Networks are 784-50-20-10 with an MSE loss function. A sigmoid non-linearity is used. A batch size of 32 is used. $B$ is updated using synthetic gradient updates with learning rate $\eta = 0.0005$, $W$ is updated with learning rate 0.0004, standard deviation of noise is 0.01. Same step size is used for feedback alignment, backpropagation and node perturbation. An initial warm-up period of 1000 iterations is used, in which the feedforward weights are frozen but the feedback weights are adjusted.

## C.2 FIGURE 3

Network has dimensions 784-200-2-200-784. Activation functions are, in order: tanh, identity, tanh, relu. MNIST input data with MSE reconstruction loss is used. A batch size of 32 was used. In this case stochastic gradient descent was used to update $B$. Values for $W$ step size, noise variance and $B$ step size were found by random hyperparameter search for each method. The denoising autoencoder used Gaussian noise with zero mean and standard deviation $\sigma = 0.3$ added to the input training data.

## C.3 FIGURE 4

Networks are 784-50-20-10 (noise variance) or 784-N-50-10 (number of neurons) solving MNIST with an MSE loss function. A sigmoid non-linearity is used. A batch size of 32 is used. Here $W$ is fixed, and $B$ is updated according to an online ridge regression least-squares solution. This was used becase it converges faster than the gradient-descent based optimization used for learning $B$ throughout the rest of the text, so is a better test of consistency. A regularization parameter of $\gamma = 0.1$ was used for the ridge regression. That is, for each update, $B^i$ was set to the exact solution of the following:

$$\hat{B}^{i+1} = \arg\min_{B} \mathbb{E} \left\| \mathbf{g}(\mathbf{h}^i, \tilde{\mathbf{e}}^{i+1}; B) - \hat{\lambda}^i \right\|_2^2 + \gamma \|B\|_F^2. \tag{18}$$

## C.4 CNN ARCHITECTURE AND IMPLEMENTATION

Code and CNN architecture are based on the direct feedback alignment implementation of Crafton et al. (2019). Specifically, for both CIFAR10 and CIFAR100, the CNN has the architecture Conv(3x3, 1x1, 32), MaxPool(3x3, 2x2), Conv(5x5, 1x1, 128), MaxPool(3x3, 2x2), Conv(5x5, 1x1, 256), MaxPool(3x3, 2x2), FC 2048, FC 2048, Softmax(10). Hyperparameters (learning rate, feedback learning rate, and perturbation noise level) were found through random search. All other parameters are the same as Crafton et al. (2019). In particular, ADAM optimizer was used, and dropout with probability 0.5 was used.

## C.5 NOISE ABLATION STUDY

The methods listed in Table 2 are implemented as follows. For the autoencoding task: Through hyperparameter search, a noise standard deviation of $c_h^* = 0.02$ was found to give optimal performance for our method. For BP(SGD), BP(ADAM), FA, the 'noise' results in the Table are obtained by adding zero-mean Gaussian noise to the activations with the same standard deviation, $c_h^*$. For the DAE, a noise standard deviation of $c_i = 0.3$ was added to the inputs of the network. Implementation of the synthetic gradient method here takes the same form as our method: $g(\mathbf{h}, \mathbf{e}, \mathbf{y}; B) = B\mathbf{e}$ (this contrasts with the form used in Jaderberg et al. (2016): $g(\mathbf{h}, \mathbf{e}, \mathbf{y}; B, c) = B^\mathsf{T}\mathbf{h} + c$). But the matrices $B$ are trained by providing true gradients $\lambda$, instead of noisy estimators based on node perturbation. This is not biologically plausible, but provides a useful baseline to determine the source of good performance. The other co-adapting baseline we investigate is the 'matching' rule (similar to (Akrout et al., 2019; Rombouts et al., 2015; Martinolli et al., 2018)): the updates to $B$ match those of $W$, and weight decay is used to drive the feedforward and feedback matrices to be similar.

For the CIFAR10 results, our hyperparameter search identified a noise standard deviation of $c_h = 0.067$ to be optimal. This was added to the activations . The synthetic gradients took the same form as above.

