# OpenReview forum: "Learning to solve the credit assignment problem"
_ICLR.cc/2020/Conference — Accept (Poster)_

### Official Review · AnonReviewer2 · 2019-10-20
**Official Blind Review #2**

**Rating:** 6

**Review:**

This paper proposes a method that addresses the "weight transport" problem [1] (not cited by the authors) emphasizing the biological infeasibility of artificial neural networks (ANNs) which are trained by gradients computed by the backpropagation algorithm [2a, 2b]. It is arguably the most eminent criticism (among many many others) when learning in the brain is modelled by such ANNs.

The authors cite only Rumelhart et al. (1986) for backpropagation (which is the same as the reverse mode of automatic differentiation), although Rumelhart cited neither Linnainmaa, the inventor of the method [2a], nor Werbos, who first applied it to ANNs [2b].

The authors propose to estimate the weights for the backward pass using a noise-based estimator and provide theoretical and empirical arguments. The proposed method is compared with backpropagation (the ground truth) and direct feedback alignment (DFA; which resorts to random matrices for the backward pass) on small fully-connected, convolutional, and fully-connected auto-encoder networks for MNIST, CIFAR10, and CIFAR100 and offers analysis based on insights from recent work.

Generally, this work is well-placed and the method straightforward and novel. That said, we found the paper to be difficult to parse with some questionable claims which we don't believe are sufficiently backed up by experiments. Due to the following issues, we think that the paper is not yet quite ready for acceptance.

1.) As the authors point out, the recent work by Akrout et al. [3] is very closely related but the paper is lacking a significant discussion let alone an empirical comparison. Extrapolating from the experiments in this paper we think that the presented method is likely going to underperform since Akrout et al. show on-par performance with backprop on imagenet.

2.) The paper appears to blend synthetic gradients [4a, 4b] and feedback alignment to a somewhat questionable degree. A connection that is usually not done in other recent papers about biologically feasible ANNs. And that for good reasons: synthetic gradients do not address the weight transport problem. They backprop the synthetic gradients in order to train the gradient estimators of the previous layer.

Also, the paper cites only the recent reference [4b] (2016) on synthetic gradients but not the original work [4a] (1990).

3.) The authors claim "Thus our method illustrates a biologically realistic way by which the brain could perform gradient descent learning". Even though their method is a more plausible way by which the brain could perform gradient descent learning, it is still far from realistic. The proposed method still implicitly "transports" a lot of information from the forward pass to the backward pass (such as a. topology or b. the derivative of the activation function or c. the activations of the forward pass (h^i) that are necessary to compute the gradient estimate in either case).

4.) The experimental results of the proposed method but also the baseline are simply too far from the state of the art. This is partially dismissed in section 5 due to regularization and data augmentation "tricks". We'd like to point out that feedback alignment has been reported to achieve <2% error on the MNIST dataset [6] which is significantly better than the results reported in this work without any tricks. Similar seems to be the case for CIFAR10 and CIFAR100. The reported results on CIFAR 10 and 100 are over 20% below the state of the art. Furthermore, many previous papers provide code which arguably makes such experiments particularly easy (see [7]). We do not expect the authors to provide state of the art results but they should be at least in the same ball-park.

5.) The authors claim "(...) this hybrid approach can solve large-scale problems." We don't think this is backed up by experiments as most models are rather small (all < 50k parameters except the CNN architecture) and evaluated on toy datasets (for vision standards; while underperforming as already mentioned).

6.) It is not obvious how the noise-based estimation will scale with width and depth. The authors make the observation that "The number of neurons has an effect". This can be seen in the appendix in Fig 4E where the relative error increases exponentially with the number of neurons in the first layer (note that the second layer doesn't increase since it remains unchanged). This seems to indicate that the method would not scale well.

7.) Why do the authors decide to train the MNIST networks with such a small learning rate (4e-4) for 2 million steps? Better results can be achieved with just a few thousand steps. Does the method work in the "general" setting?

8.) With regards to Figure 2C for layer 2: Given a random initialization of the backward pass for feedback alignment but also the presented method, we would expect that the sign congruence of the first few iterations to be the roughly the same on average. That doesn't seem to be the case according to the figure. Is there are an explanation for this observation?

9.) Why the sudden switch from feedback alignment to direct feedback alignment for the CNN experiment in section 4.3? Does the approach of section 4.2 not work with CNNs?

10) We hope the authors could clear up some confusion with regards to section 3 and the consistency proofs in the appendix.

10.a) Apparently, the notation slightly changes? In section 2, layers are indicated by 1 <= i <= N as the superscript. In section 3, i prevails but a new superscript 1 <= n <= N is used for the layers. Theorem 1 then states that the least-squares estimator for B^(i+1)\tilde{e}^{i+1} is (\hat{B}^{N+1})'. There seems to be a notation issue since it is our understanding that every layer should have its own estimated weights.

10.b) We believe A03 should state that E[\tilde{e}^i (\tilde{e}^i)'] is of full rank and not simply \tilde{e}^i (\tilde{e}^i)' assuming that \tilde{e} stands for the equivalent X of regular least-squares. From the main text (specifically the eq in section 2.1) one might assume that \tilde{e}^i are vectors but the proof seems to treat them like the design matrices X in the regular least-squares estimator.

10.c) It is awkward that in the proof of Theorem 1 and 2 the population size is suddenly T, the same symbol for the transpose.

10.d) The regular OLS estimator of the coefficients is b = (X'X)(X'y) where X is a full-rank design matrix and y are the targets. It is our understanding that \lambda, as defined by eq 12, is equivalent to the y of the regular OLS estimator. But in the following equation (the bottom of page 14), the W^{N+1} of \lambda is factored out, apparently assuming e^{N+1}e^{N+1}' = I. Maybe we made a mistake, we'd appreciate if the authors could clarify this step in the proof.

11.) Finally, we add some additional minor comments or sources of confusion:
- we'd appreciate if all equations were numerated
- In section 2, the definition of the loss is verbose and partially unnecessary. L(y,\hat{y}(x)) never appears in the text. Furthermore, \hat{y} is not introduced as a function so writing \hat{y}(x) is awkward.
- Similar for the equation in section 2.2.
- 4.1 first line: the tilde is misplaced
- Figure 3A, the black line has no label (assumed to be DAE which is missing)
- 4.2 first inline equation: the tilde is misplaced
- Table 1: no decimal values, not std. Not clear if the difference between DFA and note perturbation is significant.
- Figure 4A-E is missing the crucial comparison with feedback alignment


References:

[1] Grossberg, Stephen. "Competitive learning: From interactive activation to adaptive resonance." Cognitive science 11.1 (1987): 23-63.

[2a] Seppo Linnainmaa. The representation of the cumulative rounding error of an algorithm as a Taylor expansion of the local rounding errors. Master's Thesis, Univ. Helsinki, 1970. FORTRAN code on pages 58-60. See also BIT 16, 146-160, 1976.

[2b] Werbos, Paul J. "Applications of advances in nonlinear sensitivity analysis." System modeling and optimization. Springer, Berlin, Heidelberg, 1982. 762-770.

[3] Akrout, Mohamed MA, et al. "Deep learning without weight transport." Advances in Neural Information Processing Systems. 2019.

[4a] J.  Schmidhuber. Networks adjusting networks. In J. Kindermann and A. Linden, editors, Proceedings of `Distributed Adaptive Neural Information Processing', St. Augustin, 24.-25.5. 1989, pages 197-208. Oldenbourg, 1990. Extended version: TR FKI-125-90 (revised), Institut für Informatik, TUM. http://people.idsia.ch/~juergen/FKI-125-90ocr.pdf See section "An Approach to Local Supervised Learning in Recurrent Networks."

[4b] Jaderberg, Max, et al. "Decoupled neural interfaces using synthetic gradients." Proceedings of the 34th International Conference on Machine Learning-Volume 70. JMLR. org, 2017.

[5] Czarnecki, Wojciech Marian, et al. "Understanding synthetic gradients and decoupled neural interfaces." Proceedings of the 34th International Conference on Machine Learning-Volume 70. JMLR. org, 2017.

[6] Nøkland, Arild. "Direct feedback alignment provides learning in deep neural networks." Advances in neural information processing systems. 2016.

[7] https://paperswithcode.com/sota/image-classification-on-cifar-10?p=maxout-networks

For now, we'd lean towards rejecting this submission, but we might change our minds, provided the comments above were addressed in a satisfactory way. Let us wait for the rebuttal.

Edit: After the rebuttal, we increased our score.

**Experience Assessment:**

I have read many papers in this area.

**Review Assessment: Checking Correctness Of Derivations And Theory:**

I assessed the sensibility of the derivations and theory.

**Review Assessment: Checking Correctness Of Experiments:**

I assessed the sensibility of the experiments.

**Review Assessment: Thoroughness In Paper Reading:**

I read the paper thoroughly.

---

> ### Author Response · Authors · 2019-11-13
> **Thanks for the detailed review**
>
> Thank you for the thorough evaluation of the paper. Your comments have helped tighten the arguments and presentation of the paper.
>
> We have added additional citations to better represent the long history of research into backpropagation and weight transport.
>
> To your specific points:
>
> 1) Akrout is very closely related (as is this ICLR submission: https://openreview.net/forum?id=rJxWxxSYvB&noteId=rJxWxxSYvB). They also addresses weight transport, but they admit that they do not tackle biological plausibility. We have added these points to the discussion to better highlight the difference between the methods, in particular in their plausibility.
>
> The comparison to other co-adapting methods is indeed needed, as another reviewer noted. We offer two co-adapting baselines to understand the performance of our method. The matching method (explored in Martinolli 2018 and Rombouts 2015), and a non-noisy synthetic gradient method. Both of these in fact performance worse than our method on the cases we tested -- suggesting both gradient approximation and noise can benefit performance. See the additional results section for more details.
>
> 2) We have added additional citations on earlier work on synthetic gradients. Thanks for directing us to the 1990 Schmidhuber paper.
>
> We use the term synthetic gradient in a slightly more general sense than the exact method proposed in [1]. We use it to mean a method that approximates the gradient dL/dh (what are referred to as ‘conspiring networks’ in [2]). Reference to this terminology has been added to the text. While synthetic gradient methods as implemented in [1] do not solve the weight transport problem, the method we implement here does. To better respond to this point, we’re unclear if the reviewers feel there is something questionable in our method itself, or just in our use of the term synthetic gradients?
>
> 3.) We agree that we don’t solve all issues with efficient and biologically plausible gradient-based learning, and this statement is misleading because of this. We have changed the statement to the following.
>
> “By combining local and global feedback signals, this method provides a more plausible way the brain could solve the credit assignment problem.”
>
> To further not over-sell the method, we add some of the caveats you mention in the discussion.
>
> [1] Jaderberg, Max, et al. "Decoupled neural interfaces using synthetic gradients." Proceedings of the 34th International Conference on Machine Learning-Volume 70. JMLR. org, 2017.
> [2] Czarnecki, Wojciech Marian, et al. "Understanding synthetic gradients and decoupled neural interfaces." Proceedings of the 34th International Conference on Machine Learning-Volume 70. JMLR. org, 2017.

---

> ### Author Response · Authors · 2019-11-13
> **Issues of performance and scaling**
>
> Points 4-6 and 9 all address issues to do with scaling and performance. We address these points together.
>
> Our main interest is in understanding learning in the brain, in particular how the brain could solve the credit assignment problem. Compared with current approaches in the neuroscience literature, based largely on forms of the REINFORCE algorithm, or reward-modulated STDP, we do achieve good performance. It is in this sense that we mean ‘large-scale’ models. Details of what a more explicitly neuro model would look like can be gained by looking at, e.g., the Richards paper [3]. That is, something like their approach could be used, with the feedback weights being adjusted according to our rule, instead of being fixed as in their implementation. We have adjusted the usage in the text to be more clear about this point.
>
> Of course, while something like feedback alignment (FA) also performs on ‘large’ networks, in this sense, it only works in a very limited set of neural architectures. This work builds on a FA-like approach, extending its utility to a more diverse range of architectures and increasing its relevance to learning in the brain. We do this by combining local and global feedback signals.
>
> We believe this idea (combining local and global signals) is interesting even without state-of-the-art performance on common ML benchmarks. Ultimately, the brain must rely on some form of feedback structures to perform credit assignment. But we also know learning is modulated by ‘globally’ available feedback signals, (e.g. dopamine as a reward prediction error). It is reasonable to propose that the brain ‘solves’ the credit assignment problem using a combination of feedback networks which communicate errors specific to a given neuron, and global feedback signals available to all neurons. We show that indeed it is advantageous to do so.
>
> For the scaling, Figure 4E actually suggests that the error scales linearly with the number of neurons (both the y-axis (error) and the numbers of neurons experimented with increase linearly on a log scale).  As sign-congruence in the feedforward and feedback matrices is known to result in good performance, we believe the approach is more tolerant to the high variance in large networks that is a result of this scaling behavior. For all noise levels and numbers of neurons, Figure 4 shows an alignment with the true error signal to within around 10 degrees. This will still result in very high sign congruence, and thus good performance, for this network.
>
> We did move to DFA for the CIFAR experiment because extending the FA-based approach to the deeper and convolutional nets required for CIFAR did not provide any improvements over just fixed weights. FA is known to be difficult to work with convolutional networks, so we do not feel this is a surprising result. However, even if by itself our method does not scale to very deep networks, there are still many ways our approach is relevant to learning in the brain. For instance, recent work shows that convolutional layers of a deep network can be trained in a purely unsupervised fashion, with only the latter fully connected layers trained with a supervised learning rule, and resulting in good performance on ImageNet [4]. This approach could be combined with the method presented here, such that perturbations are only needed to solve weight transport for the final few layers. This could result in good performance on ImageNet without requiring weight transport. Our intention here is to present the method on its own, and feel such combinations are beyond the scope of the current work. These investigations are the focus of follow up studies.
>
> [3] Guergiuev J, Lillicrap T, Richards B. “Towards deep learning with segregated dendrites” eLife 2017 (6).
> [4] Grinberg L, Hopfield J, Krotov D, “Local Unsupervised Learning for Image Analysis” arXiv e-prints 2019.

---

> ### Author Response · Authors · 2019-11-13
> **Smaller comments and details of proof**
>
> 7) In general, we found in our hyperparameter search that better performance is obtained when the learning rate for the feedback weights is higher than the rate for the feedforward weights, as intuition would suggest. This ‘constraint’ appears to limit how high the feedforward learning rate can be set, at least if simply implementing SGD with a fixed learning rate. No doubt there are ways to increase the learning rate while maintaining performance. However, Figure 2 and Figure 4 are mainly intended to present the qualitative features of the methods, e.g. does it achieve good alignment, compared with fixed weights? This is a similar analysis to the original feedback alignment study, which also uses fixed rates.
>
> 8) Similar to the methods of Akrout et al 2019 [5] we implement an initial warm-up period, in which the feedforward weights are not adjusted, while the feedback weights are. This explains the behavior of the initial configurations in Figure 2C. This detail is now in the methods in the appendix.
>
> 10) a) We make the notation consistent, thanks for pointing this out. Yes, every layer does have its own feedback matrix. Theorem 1 just applies to the final layer of a deep networks (or shallow nets), hence the statement only applies to matrix B^{N+1}. The later results apply to convergence in deeper nets.
>
> b) The matrix-vector form of backprop is setup so that we can think of \tilde{e} (and h, x, y, etc) as either a vector for a single input, or matrices corresponding to a set of T inputs for a given mini-batch. For the proof, the \tilde{e} represent matrices (we could think of T as the minibatch size, and we could think of \tilde{e} as a design matrix). This point is clarified in the text.
>
> However it is not the expected value of the matrix E = \tilde{e}^i (\tilde{e}^i)' that should be full rank, but E itself. On addressing this point, we realise that we do need to amend A3. In particular, since the matrix E is a random matrix, we need to assume that it is of full rank with probability 1. We have amended A3. Thanks for drawing our attention to this point.
>
> c) We have made the transpose notation a different font to avoid confusion.
>
> d) “It is our understanding that \lambda, as defined by eq 12, is equivalent to the y of the regular OLS estimator.” Yes, this is right. We don’t assume the product is e^{N+1}e^{N+1}' = I, but rather we note that we end up with
>
> B^{N+1} = W{N+1}(e^{N+1}e^{N+1}')(e^{N+1}e^{N+1}')^{-1} + ….
>
> where the latter terms cancel and result in identity I, given assumption A3.
>
> 11.) Responses indicated with **
> - we'd appreciate if all equations were numerated
>
> ** It’s true, it may be easier to evaluate and discuss a paper with all equations numbered. However, for consistency between these posted reviews, our responses, and the initial and final submitted versions of the paper, we will keep the numbering as is, to avoid any possible confusion.
>
> - In section 2, the definition of the loss is verbose and partially unnecessary. L(y,\hat{y}(x)) never appears in the text. Furthermore, \hat{y} is not introduced as a function so writing \hat{y}(x) is awkward.
>
> ** We have removed references to \hat{y}(x).
>
> - Similar for the equation in section 2.2.
>
> ** The notation introduced in section 2.2 is used later (for example, equation (2)), so we believe it is useful to define.
>
> - 4.1 first line: the tilde is misplaced
>
> ** Fixed. Thanks
>
> - Figure 3A, the black line has no label (assumed to be DAE which is missing)
>
> ** The black line has a label in the submitted pdf we have access to (downloading from openreview, should be the same file you have). It is DAE.
>
> - 4.2 first inline equation: the tilde is misplaced
>
> ** Fixed.
>
> - Table 1: no decimal values, not std. Not clear if the difference between DFA and note perturbation is significant.
>
> ** We have added error bars
>
> - Figure 4A-E is missing the crucial comparison with feedback alignment
>
> ** Figure 4 fixes the feedforward weights. Implementing feedback alignment (fixed feedback) with fixed feedforward weights would just be implementing no learning at all. So we do not believe this is an interesting comparison.
>
> [5] Akrout, Mohamed MA, et al. "Deep learning without weight transport." Advances in Neural Information Processing Systems. 2019.

---

### Official Review · AnonReviewer3 · 2019-10-21
**Official Blind Review #3**

**Rating:** 6

**Review:**

It's unclear how multi-layer biological neural networks could implement gradient-based learning, as they don't have the symmetric connections needed for backpropagation. This paper proposes a perturbation-based synthetic gradient estimator that does not rely on symmetric backward connections. Hidden unit perturbation is used to estimate the loss gradient, and backward connections are trained via gradient descent to predict the approximate gradients from the perturbation-based estimator.

The topic is important and the paper is well-written. I don't follow this area closely, but from what I can tell it's a novel idea. The results are strong. The method beats various alternatives and closely matches backpropagation in terms of performance on the MNIST tasks (less so on CIFAR). It's especially curious that the method performs better than backpropagation on the autoencoder task.

I have an unresolved question: What do the learnable backward connections add beyond the perturbation estimator for the gradients? If the perturbation-based estimator could be used to train the forward model, does the trained backward model have advantages in terms of efficiency or performance? I would appreciate more discussion of this key choice, or a direct comparison with only the perturbation-based estimator (only) to understand the differences.

**Experience Assessment:**

I do not know much about this area.

**Review Assessment: Checking Correctness Of Derivations And Theory:**

I did not assess the derivations or theory.

**Review Assessment: Checking Correctness Of Experiments:**

I assessed the sensibility of the experiments.

**Review Assessment: Thoroughness In Paper Reading:**

I read the paper at least twice and used my best judgement in assessing the paper.

---

> ### Author Response · Authors · 2019-11-12
> **Perturbation-based methods are well explored as biologically plausible learning rules. Know that these don't scale well.**
>
> Thank you for your comments and evaluation of the work.
>
> Yes, your question is a good one. The idea of using a perturbation-only based estimate of the gradient is well explored in a number of papers, particularly by Seung and Fiete (see, for example, [1] and [2]). These methods can be seen as applying the well known REINFORCE algorithm, using noise in different parts of the circuit as perturbations. There is a connection between this theory applied in this way and reward-modulate spiking timing dependent plasticity, which makes these rules quite plausible. The issue is that these methods have only been demonstrated to work on small scale problems, nothing like the scale of models typically used in artificial neural networks trained with backpropagation. The issue is the scaling behavior: the variance of the REINFORCE estimator scales linearly with the number of neurons. A simple analysis of this is provided in [3], for example. A simple empirical demonstration provided in [4]. We mention this point in the introduction. We have also modified the discussion to make this point more clear:
>
> “By reaching comparable performance to backpropagation on MNIST, the method is able to solve larger problems than perturbation-only methods [1,2,4]. By working in cases that feedback alignment fails, the method can provide learning without weight transport in a more diverse set of network architectures. We thus believe the idea of integrating both local and global feedback signals is a promising direction towards biologically plausible learning algorithms.”
>
> References:
> [1] Xiaohui Xie and H. Sebastian Seung. Learning in neural networks by reinforcement of irregular
> Spiking. Physical Review E, 69, 2004.
> [2] Ila R Fiete, Michale S Fee, and H Sebastian Seung. Model of Birdsong Learning Based on Gradient Estimation by Dynamic Perturbation of Neural Conductances. Journal of neurophysiology, 98: 2038–2057, 2007.
> [3] Danilo Jimenez Rezende, Shakir Mohamed, and Daan Wierstra. Stochastic Backpropagation and Approximate Inference in Deep Generative Models. Proceedings of the 31st International Conference on Machine Learning, PMLR, 32(2):1278–1286, 2014.
> [4] Werfel, Justin, Xie, Xiaohui, Seung, H. Sebastian. Learning Curves for Stochastic Gradient Descent in Linear Feedforward Networks. Neural Computation 2005

---

### Official Review · AnonReviewer1 · 2019-10-23
**Official Blind Review #1**

**Rating:** 6

**Review:**

In this paper authors study one possible incarnation of more biologically plausible learning scheme, akin to (direct) feedback alignment methods, where the error signal is linearly mapped onto the update direction, without reusing forward weight (so to break the weight sharing issue, that currently is believed to be impossible for brains).
Authors approach can be summarised as a mixture of synthetic gradients with a noise-enriched estimation, rather than direct empirical risk minimisation.
Contributions are two fold, first, authors provide some theoretical analysis of the convergence of gradient estimator in a simplified setup, second, the noise-based estimator is empirically evaluated on 2 simple classification tasks.

Paper is well written, and easy to read, with relatively easy to follow notation (which is quite a tricky task for non-gradient based learning methods that require abandoning typical concept of loss minimisation and talking about dynamical systems instead).

I have a few critical comments, that I hope authors can address in the revised manuscript:
- first, high level thing, that seems to be missing from the manuscript is use of baselines that are actually co-adapted, rather than random (e.g. DFA and FA). To be more specific, in papers like "Sobolev Training for Neural Networks" (NeurIPS  2017, https://papers.nips.cc/paper/7015-sobolev-training-for-neural-networks.pdf) one can find 3 basic methods (all requiring one implementation, and differ only in terms of which loss is applied): at each layer h, the gradient predictor g(h, e, B) is composed of (d/dh) CE[softmax(Bh + c), y], which has an analytical form, is biologically plausible (as boils down to a simple affine transformation if h and y. This model can now be supervised in 3 ways:
a) one can put supervision only on the gradient (and bootstrap from higher layers, if needed, as in FA or "fully decoupled" synthetic gradient model in Jaderberg et al.) [this is essentially "pure" SG from Jaderberg et al. but constrained to the conservative vector fields, so to guarantee convergence]
b) one can put supervision on loss itself matching the "topmost loss" (reminescent of DFA, where error propagation skips entire network) [called Critic Training, Critic Network etc. depending on the source]
c) twe two above can be used jointly (which leads to the full Sobolev training)
Neither of these methods have been compared, and in reviewer's opinion it is critical, as approaches are very similar, and while Czarnecki et al. was not analysing biological plausibility, the models proposed do satisfy the same constraints requested by authors of this paper. Note, that with critic learning on MNIST, one can easily get results matching backpropagation (which, in the current manuscript is claimed as an important property of the introduced method). It might also be important to show that well tuned linear model on MNIST can reach 95% test accuracy too.
- similarly, authors are not disentangling noise-induction, from the overall setup. Which of these two is the actual source of good results? Many previous works would rely on empirical risk minimisation, if one uses noised versions and train "regular" fully decoupled affine synthetic gradients, will the results be analogous? Will this improve the Sobolev training (if implemented and verified)? Given, that both methods are known object in the literature, providing understanding of which components are actually important (maybe it is only the combination that works?) would strengthen the claims.
- Analogously, to further decouple mixed effects, when authors claim that noise-based estimator performs better than adam in the autoencoding case, having an adam with artifically added same amount of noise to gradient estimates would be helpful for the reader to see if the difference lies in the "second order estimates" of Adam, or in simply regularising effects of adding noise.  In particular, note, that this is a well established result, that noising inputs to the neural network is equivalent (up to first order terms) to Tikhonov regularisation.
- the theoretical claim of Thm 1 is quite trivial, and while I really admire authors effort to provide theoretical grounding, it feels a bit overstated in the current form. What authors are showing, is that in the simple setup, where network is not learning, and all the errors are in the sense, independent, the linear predictor is consistent. This property is well known, and both assumptions used - never met in practise, consequently I would strongly suggest downplaying the claim, and stating these results in sentence or two, with proof moved to the appendix, as in the current version of the manuscript section 3 is presented as major contribution, rather than an interesting side note.


Minor comments:
- can authors provide some error intervals for results in Table 1? Results are so close, that claiming that 48 is even "marginally better" than 47 seems odd, unless many repeated runs were conducted and some confidence intervals can be provided?
- it is odd to say "using backpropagation OR Adam". Adam is just an optimiser, it still uses backpropagation (aka chain rule). Could the notation be unified so that when talking about comparing optimisers, authors state Adam and SGD (assuming that this is what is currently called backpropagation?) and when talking about chain rule, the name of backpropagation is there?


Overall I believe it is an interesting study, however, currently missing important baselines and ablations to be a good ICLR contribution. If authors are willing to add these, I will be happy to revise the score assigned.

**Experience Assessment:**

I have published in this field for several years.

**Review Assessment: Checking Correctness Of Derivations And Theory:**

I assessed the sensibility of the derivations and theory.

**Review Assessment: Checking Correctness Of Experiments:**

I carefully checked the experiments.

**Review Assessment: Thoroughness In Paper Reading:**

I read the paper thoroughly.

---

> ### Author Response · Authors · 2019-11-15
> **We implemented baselines and noise ablations, both useful suggestions**
>
> Thank you for your evaluation and feedback. You have provided many good suggestions that, we believe, have improved the paper. We respond to you particular points below.
>
> On co-adapted baselines: This is a good idea. We have implemented a number of co-adapting methods, including the synthetic gradients and a simple matching rule, and added an additional section to the results section describing them. Note however that both the synthetic gradient and Sobolev method you suggest assume supervision from the true gradient signal, which is unrealistic in a neuro setting -- exact gradients specific to each neuron are what we’re trying to avoid. We just compare to the SG method, and not the Sobolev method. The critic training you mention is more realistic, as it just assumes supervision from the loss, this analysis was not completed by the end of the rebuttal period. Nonetheless, we have added analysis of two co-adapting methods, including the SG method you suggest, and feel this comparison has provided insight into our method. See text and below for a summary.
>
> On disentangling noise-induction: Also a good idea. In the additional results section we discuss the effect that noise has on the other models. In short, we find that by itself adding noise to SGD training with backprop increases performance (both noise in inputs and activations) on the autoencoder but not on CIFAR task. Adding noise to SGD training with feedback alignment does not help on the autoencoder task. This suggests that our method is benefiting not just from the addition of noise but also from learning a less biased approximation of the gradients; noise doesn’t always help, using a less-based gradient estimator seems to always help.
>
> On decoupling mixed effects: Yes, also a good point. To focus the discussion, we have mostly used SGD training with different gradient estimators. We keep the ADAM results in, but do not discuss them at length, making sure to separate the gradient estimation from the optimization method used.
>
> On the theoretical results: We agree the result of Theorem 1 is not so surprising. However, the follow-up results -- that the estimators are consistent through-out the rest of the layers, for both linear ‘feedback’ and ‘direct feedback alignment’-type estimators -- we feel are not as trivial, and worthy of mention in the main text. But your point is taken: To make space for the new empirical analysis, we have reduced the size of the theory section with shorter ‘informal’ theorem statements, with all the details now in the supplement.
>
> On the minor comments:
> * We have added error intervals for the results in Table 1.
> * We have been more careful to refer to ADAM as the optimizer, in contrast to the gradient approximation method, as you have suggested. Thanks.

---

### Author Response · Authors · 2019-11-15
**Response to reviewers**

We would like to thank the reviewers for their comments and helpful suggestions. We have updated the manuscript, and believe we have addressed the concerns that were raised.

---

### Decision · Program_Chairs · 2019-12-19

**Decision:**

Accept (Poster)

**Comment:**

Initial reviews of this paper cited some concerns about a lack of comparison to SOTA and baselines, and also some debate over claims of what is (or is not) "biologically plausible."  However, after extensive back-and-forth between the authors and reviewers these issues have been addressed and the paper has been improved.  There is now consensus among authors that this paper should be accepted.  I would like to thank the reviewers and authors for taking the time to thoroughly discuss this paper.